# Associations Between Regulatory Immune Cells, Thymus Cellular Remodeling, and Vascular Aging in Advanced Coronary Atherosclerosis: A Pilot Study

**DOI:** 10.3390/diagnostics15192494

**Published:** 2025-09-30

**Authors:** Irina Kologrivova, Alexey Dmitriukov, Natalia Naryzhnaya, Olga Koshelskaya, Olga Kharitonova, Alexandra Vyrostkova, Elena Kravchenko, Ivan Stepanov, Sergey Andreev, Vladimir Evtushenko, Anna Gusakova, Oksana Ogurkova, Tatiana Suslova

**Affiliations:** Cardiology Research Institute, Tomsk National Research Medical Center, Russian Academy of Sciences, 111A Kievskaya, Tomsk 634012, Russia; aldmn9k@mail.ru (A.D.); natalynar@yandex.ru (N.N.); koshel@live.ru (O.K.); hoa@cardio-tomsk.ru (O.K.); alexandra.vy20@gmail.com (A.V.); nikonova@cardio-tomsk.ru (E.K.); i_v_stepanov@mail.ru (I.S.); anselen@rambler.ru (S.A.); vv_e@cardio-tomsk.ru (V.E.); anna@cardio-tomsk.ru (A.G.); oon@cardio-tomsk.ru (O.O.); tes@cardio-tomsk.ru (T.S.)

**Keywords:** aging, atherosclerosis, Gensini score, vascular aging index, myeloid-derived suppressor cells, FoxP3+ T regulatory lymphocytes, thymus, endothelial precursor cells, senescence-associated secretory pattern

## Abstract

**Background/Objectives:** Biological aging phenotypes in coronary artery disease (CAD) include coronary atherosclerosis, vascular aging, and endothelial dysfunction. The aim of the present study was to investigate the potential links between aging phenotypes, regulatory immune cells, and features of the thymus in patients with CAD. **Methods:** A single-center, cross-sectional, comparative study was conducted. Patients were stratified according to the severity of coronary atherosclerosis: patients with a Gensini score ≥ 65 points and patients with a Gensini score < 65 points. Peripheral blood and thymus biopsy were obtained. Imaging flow cytometry, ELISA, and immunohistochemical analysis were used for analysis. **Results:** Thymic morphology ranged from total fatty involution to a preserved structure of the thymus (20–80% area in 31% of obtained samples) but was not associated with the severity of atherosclerosis. Meanwhile, patients with a Gensini score ≥ 65 had impaired thymus cellular composition compared to patients with a Gensini score < 65 points; increased frequency of CD8+ T lymphocytes and NK cells; and decreased frequency of CD4 + CD8+ T lymphocytes. In peripheral blood, the main determinants of a Gensini score ≥ 65 points were low absolute counts of eMDSCs and CD25^low^ Tregs with FoxP3 nuclear translocation, while advanced vascular aging was associated with elevated eMDSC absolute counts. Advanced coronary atherosclerosis was also associated with decreased numbers of endothelial progenitor cells in circulation. **Conclusions:** Thymus dysfunction accompanies CAD progression and is manifested in changes in cellular composition rather than morphology. In CAD patients, MDSC and Treg lymphocytes are equally involved in the progression of coronary atherosclerosis, which is aggravated by the decreased regulatory potential of the endothelium. Vascular aging represents a distinct phenotype of biological aging in CAD patients, characterized by the expansion of eMDSCs.

## 1. Introduction

Chronic low-grade inflammation has been attributed to the pathogenesis of cardiovascular disorders associated with atherosclerosis for several decades [1]. Meanwhile, it has been acknowledged that low-grade inflammation is also associated with biological aging, and the term “inflammaging” has been coined [2].

In coronary artery disease (CAD), biological aging at the cellular level may manifest in the development of such separate but closely related phenotypes as atherosclerosis, vascular aging, and endothelial dysfunction [3,4]. One might follow the timeline to discover which structural and mechanical changes in blood vessels result in the formation of plaque [5]. Most often, a mixed vascular aging phenotype develops (stiff vessels and atherosclerotic plaques). However, vascular remodeling may persist as it is, even in the absence of atherosclerosis. Then, vessels undergo remodeling without the formation of plaques [4,6]. A vascular aging index, based on the “intima–media” complex thickness and carotid–femoral pulse wave velocity, has been proposed for quantitative evaluation of vascular aging [7,8]. The identification of pathological factors that determine the increase in arterial stiffness and vascular aging and lead to its progression to atherosclerosis may help in prognosis, the prevention of unfavorable cardiovascular events, and targeting organ damage [9].

One of the possible drivers of accelerated aging during CAD is the failure of anti-inflammatory protective mechanisms at several levels [10]. The main cells that control the degree of inflammatory response are myeloid-derived suppressor cells (MDSCs) and T regulatory lymphocytes (Tregs) [10,11,12]. MDSCs arise in the bone marrow as a result of augmented myelopoiesis and represent immature forms of myeloid cells with immune-suppressive properties. Though initially protective, MDSCs may later aggravate inflammaging, as the clearance of cellular degradation products may be impaired [10,13]. Currently, three subsets of MDSCs are distinguished in humans: polymorphonuclear (PMN-MDSCs), monocytic (M-MDSCs), and early-stage MDSCs (eMDSCs) [14]. One of the mechanisms through which MDSCs may induce immune suppression is their ability to activate Tregs [10]. Tregs in turn are lymphoid cells with immunosuppressive activity, which may arise either in the thymus (thymic Tregs) or differentiate from the naïve T lymphocytes in the periphery (peripheral Tregs). Tregs undergo both qualitative and quantitative changes in aging [15]. Evaluation of MDSC subsets and Tregs in relation to phenotypes of biological aging in CAD may shed light on the pathogenesis of vascular aging and atherosclerosis development and identify potential diagnostic and therapeutic targets.

Aging is also associated with thymus involution and replacement of the functional thymus with adipose tissue. It can favor the development of atherosclerosis in several ways: a decrease in thymic Treg production, a decline in naïve T lymphocyte output, the constriction of T cell receptor (TCR) repertoire diversity, and the appearance of potentially auto-aggressive T lymphocytes, due to the decline in negative selection in the thymus [16,17]. Data on the interconnection between the thymus, the systemic regulatory capacity of the immune response, and inflammation with respect to biological aging in atherosclerosis remain very scarce. For a long time, the thymus in elderly patients has been regarded as merely a retrosternal accumulation of fat. However, new findings indicate that even at advanced age, the thymus retains an islet of its original structure containing thymocytes and thymic epithelial cells [18]. Whether this structural integrity is beneficial or harmful to the progression of vascular aging and atherosclerosis remains unexplored. The cellular composition of the intact thymus tissue in elderly patients with CAD is not described.

Hence, the aim of the present study was to investigate the potential correlation between various phenotypes of aging (the degree of atherosclerosis, vascular aging, and endothelial dysfunction), the cellular composition and morphology of the thymus, and the properties of immune cells with regulatory properties in patients with CAD.

## 2. Materials and Methods

### 2.1. Patients

A single-center, cross-sectional, comparative study was conducted in accordance with the latest edition of the Declaration of Helsinki and the “Rules of Clinical Practice in the Russian Federation”, approved by the Order of the Ministry of Health of the Russian Federation (19 June 2003, No. 266). Approval from the Biomedical Ethics Committee of the Cardiology Research Institute, Tomsk NRMC, was obtained (Protocol No. 210 from 18 February 2021). We recruited 56 patients with CAD scheduled for selective coronary angiography. All participants recruited for the study provided their informed consent.

The severity of atherosclerosis was evaluated via calculation of the Gensini score (GS) based on the results of selective coronary angiography [18,19]. According to our previous study, a GS value equal to 65 points (the upper quartile of the total sample) allows the identification of CAD patients with the most severe coronary atherosclerosis [20]. Thus, we stratified the patients into two groups: group 1, with GS < 65 points (*n* = 37), and group 2, with GS ≥ 65 points (*n* = 19).

Exclusion criteria were as follows: acute vascular complications less than 6 months before the study (including acute cerebrovascular events, acute coronary syndrome, or acute myocardial infarction); confirmed symptomatic arterial hypertension; any other severe comorbidity (renal failure, liver failure, autoimmune disorders, or cancer); acute inflammatory disease other than atherosclerosis at least 1 month prior to the study; refusal to participate in the study.

Anthropometry included evaluation of the body mass index (BMI) and measurement of waist circumference.

The vascular aging index (VAI) was calculated based on the results of B-mode ultrasound scanning of carotid arteries (ultrasound diagnostic system “ACUSON” 128 XP/10, Mountain View, CA, USA) and the results of oscillometric arteriography (TensioMed, Budapest, Hungary) using the following formula [7,8]:
VAI = (ln (1.09) × 10 IMT+ ln (1.14) × cfPWV) × 39.1 + 4.76,
where IMT—medium intima–media thickness of carotid arteries (mm);

cfPWV—carotid–femoral pulse wave velocity (m/s).

### 2.2. Thymus Biopsy

Samples of thymus tissues were obtained from 28 CAD patients during coronary artery bypass grafting (CABG) in the amount of 0.2–1 g, as previously described in [21]. Half of the explanted tissue was placed in pre-warmed M199 medium (37 °C), and the other half was placed in 10% buffered formalin.

The samples in M199 medium were delivered to the laboratory, where stromal vascular fraction (SVF) was extracted. Briefly, adipose tissue was minced with scissors. After that, a 5 mL solution of 1 mg/mL collagenase type I (PanEco, Moscow, Russia) in Krebs–Ringer buffer was added. The pre-warmed Krebs–Ringer buffer was added to the samples in a 1:1 ratio to terminate the reaction after 35–40 min of incubation. The cell suspension was filtered through the nylon mesh (Falcon™Cell strainer, Fisher Scientific, Waltham, MA, USA, 100 μm); thymic adipocytes were aspirated, followed by the subsequent filtering of the remaining SVF cell suspension through a 70 μm nylon mesh (Falcon™Cell strainer, Fisher Scientific, Waltham, MA, USA). The cells were then centrifuged (400× *g*) and washed twice with Krebs–Ringer buffer. Finally, the cells were resuspended in 1 mL of RPMI 1640 medium and stained for flow cytometry analysis.

The samples were fixed in formalin, maintained for 24 h, and then embedded in paraffin.

### 2.3. Blood Processing

Fasting peripheral venous blood was collected in EDTA, heparinized tubes, and tubes with an activator of clotting within a 5-day interval after coronary angiography and prior to CABG. The clotting activator tubes were centrifuged at 1500× *g* for 15 min to obtain blood serum. The serum was aliquoted and stored at −40 °C until final analysis.

EDTA and heparinized blood was used for isolation of peripheral blood mononuclear cells (PBMCs). Blood was dissolved in phosphate-buffered saline (PBS) in a 1:1 ratio and layered on Histopaque 1077 (Sigma, Livonia, MI, USA), followed by centrifugation at 400× *g*. The buffy coat was collected and washed twice with PBS. The cell suspension was processed for flow cytometry analysis.

### 2.4. Imaging Flow Cytometry

Subsets of Tregs were identified both in thymic SVF and in PBMCs. MDSCs and circulating EPC subsets were identified in the PBMC fraction. T-, B-, and NK cells were analyzed in thymic SVF. The following antibodies conjugated with fluorochromes were used to stain antigens localized on the outer cellular membrane: anti-CD4-FITC, anti-CD25-PE, anti-CD45-APC-Cy7, anti-lineage-FITC (CD3, CD14, CD16, CD19, CD20, CD56), anti-CD33-PE, anti-CD15-PE-CF594, anti-HLA-DR-PE-Cy5, anti-CD14-AF-647, anti-CD11b-APC-Cy7, anti-CD45-FITC, anti-CD34-PE, anti-CD31-PE-Cy7, anti-CD133-APC, and TBNK-cocktail (anti-CD3-FITC, anti-CD16/CD56-PE, anti-CD45-PerCP-Cy5.5, anti-CD4-PE-Cy7, anti-CD19-APC, anti-CD8-APC-Cy7) (all reagents: BD Biosciences, San Jose, CA, USA).

In the Treg protocol, cells were fixed after the staining of surface antigens, permeabilized with a specialized buffer set (BD Biosciences, San Jose, CA, USA), stained with anti-FoxP3-AF647, and then fixed and stained with the DNA dye 7-aminoactinomycin D (AAD).

The cells were analyzed using the Amnis FlowSight (Cytek Biosciences, Fremont, CA, USA) equipped with 488 nm and 642 nm lasers. Brightfield images were acquired in channel 1. A 785 nm laser was used for evaluation of side-scatter in channel 6. Single-color stained controls were used to create a compensation matrix, which was adjusted manually during the analysis. Data analysis was performed in IDEAS 6.2.64.0 software (Amnis Corporation, Seattle, DC, USA). The percentage of cells in the target population was calculated. For Treg lymphocytes, the percentage of CD25^hi^ and CD25^low^ FoxP3+ Tregs was estimated. Also, the percentage of Treg lymphocytes with FoxP3 nuclear translocation was assessed using the feature “Similarity Morphology” in IDEAS 6.2.64.0 software. CD4 + FoxP3 + CD31+ cells were regarded as Tregs of recent thymic origin (Figure 1). 

### 2.5. Enzyme-Linked Immunosorbent Assay (ELISA)

The concentrations of TNF, IL-1β, IL-6, IL-10 (all cytokine kits—VECTOR-BEST, Novosibirsk, Russia), transforming growth factor-β (TGF-β) (Invitrogen, Vienna, Austria), insulin-like growth factor (IGF) (Mediagnost, Reutlingen, Germany), endothelin-1 (Sigma, Darmstadt, Germany), resistin (Mediagnost, Reutlingen, Germany), and sortilin (AVISCERA BIOSCIENCE, Santa Clara, CA, USA) were assessed in serum by ELISA using the instrument Infinite F500 (Tecan, Männedorf, Switzerland).

### 2.6. Biochemical Analysis

The concentration of homocysteine in serum was assessed by the enzymatic cycling method (DiaSys, Holzheim, Germany) using a semi-automatic spectrophotometer for clinical biochemistry Clima MC-15 (RAL Técnica para el Laboratorio, S.A., Barcelona, Spain).

### 2.7. Multiplex Analysis of Cardiovascular Biomarkers

Multiplex analysis was performed at the Core Facility “Medical genomics”, Tomsk NRMC. The MILLIPLEX map Human Cardiovascular Panel was used to detect endocan-1 in blood serum using the Multiplex Instrument FLEXMAP 3D from the Luminex Corporation. MILLIPLEX Analyst 5.1 software (Merck KGaA, Milliplex; Darmshdadt, Germany) was used for data analysis.

### 2.8. Immunohistochemistry

Sections of paraffin-embedded thymus tissue, measuring 2–3 μm, were dried at 60 °C overnight and then placed in the automatic staining system Leica BOND-MAX (Leica Microsystems GmbH, Wetzlar, Germany). The sections were deparaffinized and antigen retrieval was performed in pH 8.8 EDTA-buffer (98 °C, 20 min). A peroxidase block was applied for 10 min after rinsing of the slides (Bond Polymer Refine Detection Kit DC9800 (Leica Microsystems GmbH). The sections were rinsed and incubated with primary anti-FoxP3 antibody (EP340, Cell Marque, Rocklin, CA, USA) for 10 min, followed by incubation with polymer for 10 min. Finally, the sections were treated with DAB chromogen for 10 min and stained with hematoxylin and eosin for 10 min. FoxP3 expression was evaluated as the number of positively stained cells in 10 fields of vision (×400 magnification). Staining was considered positive when FoxP3+ cells were detected. The percentage of the preserved thymus tissue was evaluated with respect to the total section area.

### 2.9. Statistical Analysis

The Shapiro–Wilk test was used to assess data distribution. Continuous variables were presented as the median (Me) and interquartile intervals (Q1; Q3). Discrete variables were presented as counts (*n*) and percentages (%) in the sample. Differences between independent samples were assessed using the Mann–Whitney U test (for continuous variables) and Pearson’s χ^2^ test or Fisher’s Exact test (for discrete variables). The presence of correlations between different parameters was evaluated by the Spearman correlation coefficient. The prognostic significance of the variables for the prediction of severe atherosclerosis or advanced vascular aging was determined by multiple logistic regression. In the tests, a *p*-level < 0.05 was considered to be statistically significant. All calculations were performed in Statistica 10.0 (StatSoft Inc., Tulsa, OK, USA).

## 3. Results

### 3.1. Baseline Characteristics of Patients

The baseline characteristics of the patients are presented in Table 1. Patients with GS ≥ 65 more often had myocardial infarction in their medical history. Other parameters were comparable between the groups (Table 1). The vascular aging index was significantly higher than chronological age in both groups of patients (*p* < 0.001) (Table 1).

### 3.2. Immune Regulatory Cells in Peripheral Blood

The gating strategy of MDSC subsets is represented in Figure 2.

The absolute counts of eMDSCs were lower in patients with GS ≥ 65 compared to patients with GS < 65 (Table 2). Patients with GS ≥ 65 also had decreased absolute counts of CD25^hi^ and CD25^low^ Tregs and absolute counts of Tregs with FoxP3 nuclear translocation (Table 2).

### 3.3. Circulating Endothelial Precursor Cells in Peripheral Blood

Patients with the most advanced atherosclerosis were characterized by a decreased percentage of EPCs in the CD34+ cell population and tended to have lower absolute EPC counts and a lower percentage of EPCs in the total fraction of PBMCs (Figure 3).

### 3.4. Immune Composition of Atrophied Thymus

We performed IHC analysis of 22 thymus biopsies. In seven (31%) samples, the preserved integrity of the thymus (10–80% of intact tissue) was observed (Figure 4B; Table 3). We did not reveal any interconnection between the preservation of thymic morphology and the severity of atherosclerosis (Table 3). However, only two out of seven patients with a preserved thymus had a VAI above the median value (83.9 points). The median VAI of the remaining patients with retained thymic morphology constituted 73.9 points, which was significantly lower than that in the total group of patients (*p* = 0.005).

The frequency of FoxP3+ staining in the thymus positively correlated with the area of the preserved thymus tissue in patients with atherosclerosis (r_s_ = 0.614; *p* = 0.003).

Next, we analyzed the cellular composition of thymic SVF by flow cytometry. Patients with GS ≥ 65 had more CD8+ T-lymphocytes and NK cells and a lower percentage of double-positive CD4 + CD8+ T-lymphocytes compared to patients with GS < 65 (Table 4). A more detailed analysis of the CD4 + CD8+ subset revealed that this difference was predominantly restricted to the decrease in the potentially regulatory CD4^hi^CD8^low^ subset (Table 4).

Even though patients with GS < 65 had a higher percentage of Tregs with FoxP3 nuclear translocation in the thymus, this difference was not statistically significant (Table 4).

Solely in patients with GS < 65, we revealed the presence of CD31+ early thymic Tregs (Table 4). In patients with the most advanced atherosclerosis (GS ≥ 65), CD31+ Treg cells were absent in the thymus. Thus, Tregs detected in the thymus in this group of patients most probably represent recirculated cells from the periphery (Table 4).

We revealed multiple correlations between cellular subsets in the thymus and counts of cells in peripheral blood (Figure 5). The eMDSC numbers were associated with the percentage of CD25^hi^FoxP3 + CD31+ cells in the thymus, while the percentage of CD25^low^ cells with FoxP3 nuclear translocation in the thymus positively correlated with the number of EPCs in the peripheral blood. Other than that, M-MDSCs positively correlated with inflammatory NK cells and negatively correlated with the regulatory CD4^hi^CD8^low^ subset in the thymus. Negative correlations were revealed between CD25^low^ cells with FoxP3 nuclear translocation in the blood and the frequency of CD25^low^FoxP3+ cells in the thymus (Figure 5).

### 3.5. Senescence Associated Molecular Pattern in Peripheral Blood

We did not reveal any significant differences in the concentrations of serum molecules associated with senescence (SASP molecules) or markers of endothelial dysfunction between groups of patients with varying degrees of atherosclerosis (Table 5).

Meanwhile, multiple correlations were revealed between SASP markers and cells in blood and thymus, as well as with the Gensini score. In the total group of patients, the percentage and absolute counts of eMDSCs inversely correlated with the concentration of endothelin (r_s_ = −0.480; *p* = 0.038 and r_s_ = −0.497; *p* = 0.031, respectively). The percentage of EPCs, as well as EPC absolute counts, correlated with the concentration of IGF (r_s_ = 0.549; *p* = 0.015 and r_s_ = 0.533; *p* = 0.019). The frequency of CD25^hi^ Tregs in peripheral blood inversely correlated with the concentration of TNF (r_s_ = −0.400; *p* = 0.023), while absolute counts of CD25^hi^ Tregs with FoxP3+ nuclear translocation inversely correlated with the concentration of IL-1β (r_s_ = −0.352; *p* = 0.048). The numbers of CD25^low^ Tregs in the thymus positively correlated with the concentration of homocysteine (r_s_ = 0.738; *p* = 0.037). The GS values positively correlated with the concentration of TNF (r_s_ = 0.359; *p* = 0.016).

### 3.6. Degree of Atherosclerosis and Cells of Immune System in Patients with CAD

We created a model of multiple logistic regression describing the contribution of immune cells to the severity of atherosclerosis (*p* = 8.5986 × 10^−5^; Table 6). The history of myocardial infarction was the main predictor for patients to have the most severe and wide-spread atherosclerosis. Absolute counts of eMDSCs and CD25^low^ Tregs with FoxP3 nuclear translocation significantly improved the characteristics of the model, even though each variable in particular had a *p*-level below statistical significance (Table 6).

The ROC-curve describing the model is presented in Figure 6. The accuracy of the model was equal to 75.4%; specificity was 77.1%; and sensitivity was 72.2%.

### 3.7. Vascular Aging Index and Immune Cells in Patients with CAD

We analyzed whether VAI is associated with any of the cellular or molecular indicators. In the total group of patients, the VAI correlated with the frequency of NKT cells in the thymus (r_s_ = 0.664; *p* = 0.018) and concentration of homocysteine (r_s_ = 0.351; *p* = 0.017).

The subgroup analysis demonstrated that in patients with GS < 65, the VAI inversely correlated with the frequency of double-positive CD4 + CD8+ T-lymphocytes in the thymus (r_s_ = −0.833; *p* = 0.010). In this group of patients, the concentration of IL-1β also tended to correlate with the VAI (r_s_ = 0.303; *p* = 0.073).

Meanwhile, in patients with the most advanced atherosclerosis (GS ≥ 65), the VAI correlated with the frequency of eMDSCs (r_s_ = 0.564; *p* = 0.028) and their absolute counts in peripheral blood (r_s_ = 0.682; *p* = 0.005).

We created a model of logistic regression according to which chronological age and absolute count of eMDSCs allowed the prediction of the increase in the VAI above its median value (83.9 points). The characteristics of the model are presented in Table 7 (*p* = 0.008). Even though the chronological age was not a significant determinant of the VAI increase (quite surprisingly), it increased the total predictive power of the model.

Of note, absolute counts of eMDSCs had a negative effect on the development of severe atherosclerosis, and a positive effect on the VAI (Table 6 and Table 7).

The ROC-curve describing the model had area under the curve of 0.718 (Figure 7), its sensitivity was 77.4%; and its specificity was 65.4%.

## 4. Discussion

In our work, we have demonstrated that regulatory cells from both adaptive immunity (Treg lymphocytes) and innate immunity (MDSC) are involved in the manifestation of various phenotypes of aging in CAD patients. Moreover, the immune composition of the thymus in aging patients but not its structural integrity is associated with the degree of atherosclerosis.

Data on the role of the thymus in CAD are very scarce. Even though the thymus undergoes age-related involution from early childhood, it preserves the islets of the functional tissue until the most advanced age [22]. A decline in thymus function evaluated by the number of signal-joint T cell receptor excision circles (sj-TRECs) was associated with more advanced atherosclerosis and the development of acute coronary syndrome [23]. Normally, potentially self-reactive T lymphocytes are eliminated through apoptosis in the process of negative selection [24]. However, during the constriction of the thymus and its replacement by adipose tissue, auto-reactive T cell clones may escape to the periphery, including those targeting apolipoproteins or antigens of the vascular vessels’ wall [25]. The diminished production of Treg lymphocytes represents another mechanism through which the thymus may be interconnected with atherosclerosis [26,27].

According to our data, the structural involution of the thymus was associated with the biological aging of arteries only to a certain extent: the majority of patients with a preserved thymus were characterized by lower values of VAI compared to the total group of patients. Meanwhile, the cellular composition of thymus tissue was primarily linked both to the degree of atherosclerosis and to the intensity of vascular aging. In particular, an excessive quantity of cytotoxic lymphocytes (CD8+ and NK cells) and decreased numbers of potentially regulatory CD4 + CD8+ T lymphocytes [28] were associated with the most pronounced and severe atherosclerosis in CAD patients. Vascular aging in turn was associated with increased counts of NKT cells and, similarly, with decreased numbers of CD4 + CD8+ T lymphocytes. The presence of inflammatory immune cells in the normal thymus has been confirmed in previous studies and extensively reviewed elsewhere [29]. However, the role of these cells partially remains elusive, and data on their function in atrophied thymus in the absence of thymic epithelial cells are absent. One cannot exclude the possibility of inflammatory cells’ transmigration to the periphery, where they can modulate the development and progression of vascular aging and atherosclerosis.

NK cells may promote atherosclerosis via IFN-γ production or activate CD4+-dependent mechanisms, while CD8+ lymphocytes can enhance the spread and vulnerability of plaques through the production of perforin and granzyme B, thus destroying altered macrophages and endothelial and smooth muscle cells in the blood vessels [30]. Even though the predominant expansion of cytotoxic cell lineages in the thymus is likely to support the viral theory of atherosclerosis, their subsequent accumulation in atherosclerotic lesions may be antigen-independent [31]. It has been demonstrated that CD8+ T cells are characterized by a compromised phenotype in aging, promoting inflammaging, while the response to newly encountered antigens is aggravated [18]. In addition, CD8+ T lymphocytes in aging are enriched with self-reactive cells [32]. It is still unclear if CD8+-associated impairments originate from the thymus, or are dependent upon the microenvironment.

Unfortunately, we were not able to evaluate the origin of cytotoxic cells in the thymus and therefore cannot exclude that they represent recirculated cells. Both activated CD4+ and CD8+ may return to the thymus, with CD4+ performing a regulatory function and, possibly, promoting thymus involution [33]. We did not find any evidence of the ability of NK cells to recirculate in the thymus, but such a possibility exists, as they express chemokine receptors required for thymus homing [34,35]. The investigation of transmigration of T lymphocytes and NK cells between the thymus and peripheral tissues during atherosclerosis represents a perspective scope of future studies.

Moreover, we revealed that all Tregs in the thymus of patients with the most advanced atherosclerosis did not express the CD31 molecule, a marker of recent thymic emigrants (RTE) [36]. It implies that thymic Tregs, not expressing CD31, represent a subset of cells which have undergone recirculation and returned to the thymus from the periphery [37,38]. These cells suppress the production of thymic Tregs de novo [39], which further undermines the regulatory capacity of the immune system. In the future, our findings may be further corroborated by expansion of the biomarker panel used to identify the RTE Treg subset. Currently, a universal approach for identification of this cellular subset is absent. Other surface (protein-tyrosine kinase 7 (PTK7), CD38) or intracellular (T cell receptor excision circles (TRECs), IL-8) markers may be used to verify the absence of newly produced Tregs in the most advanced coronary atherosclerosis [40].

The consequences of the exhaustion of central immune tolerance may include impairments in peripheral Treg cells in patients with the most advanced and wide-spread atherosclerosis. We observed a reduction in FoxP3+ Treg absolute counts and decreased FoxP3 nuclear translocation in patients with GS ≥ 65. In our study, we did not evaluate the functions of Tregs, but evidence from the previous experimental works indicates that FoxP3 subcellular localization strongly correlates with the immunosuppressive capacity of Tregs. Thus, Magg T. et al. (2012) demonstrated that FoxP3 nuclear translocation is associated with increased transcription of molecules involved in the regulation of Tregs (IL-2 receptor, CCR4), downregulation of inflammatory molecules (IL-2, IL-4, TNF), and an increased ability of Tregs to suppress the proliferation of CD3+ lymphocytes [41]. Ni X. et al. (2019) also confirmed that perinuclear localization of FoxP3 due to the decline in its ubiquitination is associated with the failure of Tregs to suppress the production of IL-17 and IFN-γ by CD3+ T cells [42]. Small numbers of thymic Tregs available for analysis in elderly patients significantly impedes the performance of functional assays. Previously, we demonstrated that the total fraction of thymic stromal–vascular fraction cells in patients with the most advanced atherosclerosis was prone to an increase in pro-inflammatory cytokine secretion compared to that from patients with a less pronounced atherosclerotic burden, which inversely correlated with the percentage of thymic Tregs with nuclear FoxP3 [43].

In our study, changes in the adaptive compartment of immune regulation correlated with the reduction in eMDSC absolute counts. The suppressive capacity of eMDSCs exceeds that of M-MDSCs or PMN-MDSCs [44]. The correlations between Tregs and eMDSCs have been demonstrated in various pathologies [45,46,47]. Most likely, MDSCs may influence Tregs through the production of cytokines, IFN-γ, and IL-10 [45]. Also, eMDSCs may affect the epithelial–mesenchymal transition (EMT), which may be involved in adipose thymus involution [48]. Thus, the possibility that eMDSCs may affect Tregs at the central level in the course of severe atherosclerosis cannot be excluded. However, the opposite is also possible: Treg-derived TGF-β appeared to be indispensable for MDSC generation [49]. In addition, MDSCs and Tregs share mutual signaling pathways [50].

On the other hand, EPCs were also interconnected with the percentage of thymic CD25^low^ cells with FoxP3 nuclear translocation. FoxP3+ Tregs appear to play an important role in the maintenance of bone marrow hematopoiesis [51,52]. Thymus involution precedes the decline in bone marrow function in elderly patients [53], and the inability of bone marrow to replenish the loss of vascular cells during compromised arterial integrity is associated with the progression of atherosclerosis [54]. Apparently, the revealed EPC and eMDSC deficiency in patients with the most advanced atherosclerosis is associated with the common defects in bone marrow progenitors and may be interconnected with impaired Treg production in the thymus.

Inflammaging may be manifested through SASP, comprising molecules with sometimes mutually exclusive functions [55]. We did not reveal any differences in the serum concentration of SASP markers between the groups. It may explain the high severity of atherosclerosis in both groups, as all the patients were characterized by a severity of stenosis sufficient to be scheduled for CABG surgery. Also, serum SASP concentrations might not be very representative for the local processes within the plaque, as many SASP effects are produced through paracrine interactions [56]. Meanwhile, in our study, certain SASP cytokines and growth factors were interconnected with regulatory cell mobilization and endothelial regenerative potential. IGF-1 is described as a potent anti-aging molecule and activator of PI3K-AKT/mTORC1 signaling pathways [57]. It can affect EPC mobilization [57], which may explain the direct interconnection between IGF and EPC numbers revealed in our study. The possible mechanisms may involve a signaling pathway mediated by endothelial NO synthase [58]. The frequency and characteristics of FoxP3+ Tregs appeared to be dependent upon the inflammatory cytokines TNF and IL-1β (as expected), as well as upon the non-proteinogenic amino acid homocysteine, representing an intermediate product of methionine metabolism [59]. Even though there are data on the interconnection between homocysteine and the activation of conventional T lymphocytes [59,60], data on their involvement in Treg metabolism and function are not very straightforward and require further investigation.

As expected, the advanced values of the Gensini score in CAD patients were associated with a history of myocardial infarction, while biological age was one of the predictors of high VAI values. However, we also demonstrated that immune mechanisms are involved in the development of both phenotypes of biological aging, and what is more, the role of immune cells in the development of advanced atherosclerosis and vascular aging may differ.

The presence of GS ≥ 65 points was dependent upon the decreased counts of eMDSCs and the level of FoxP3 nuclear translocation in terminally differentiated Tregs, while increased eMDSC counts were associated with exaggerated vascular aging. MDSCs represent the main cellular subset that is elevated in aging bone marrow, depending on the NF-κB signaling pathway [61]. They may mediate either immune suppression through the production of cytokines [10,45], small molecules, like NO [62], or cell-to-cell contact [63,64]. MDSCs are also capable of inflammatory activity in specific milieu [65]. Data on the functional activity of MDSCs in aging are controversial: some authors state that aging does not impair MDSC functions [61], while others indicate increased MDSC functional activity in elderly patients [66]. This discrepancy probably arises from the fact that only cell-to-cell interactions are aggravated in the elderly, while NO generation remains stable. The possible explanation of opposite effects of MDSCs in vascular aging and atherosclerosis, according to our models of multiple logistic regression, may be bone marrow exhaustion in the most advanced atherosclerosis and the inability of eMDSCs to suppress exaggerated inflammation in the vascular wall. The level of FoxP3 nuclear translocation increased the predictive capacity of our model for advanced atherosclerosis. Thus, Treg–MDSC crosstalk [45,49] may be crucial for atheroprotection and is required to decrease the degree of atherosclerosis in CAD patients.

In our study, the VAI was derived from cfPWV and intima–media thickness values. However, evaluation of coronary artery calcium (CAC) might improve the evaluation of vascular age in the future. High CAC significantly correlated with arterial stiffness and appeared to be an efficient instrument for risk stratification in CAD patients [67,68].

The main limitations of the study are its cross-sectional design and relatively low number of patients enrolled, especially for thymus biopsy. The cross-sectional design did not allow distinguishing “cause-and-effect” relationships between vascular aging, thymus cellular composition, and circulating immune regulatory cells. Immune dysregulation may either precede or be the consequence of vascular aging and atherosclerosis. We regard this work as a pilot study. Prospective and large-scale studies, even of multi-center design, will allow verification of the prognostic significance of the identified factors and provide statistically significant results. We also acknowledge the presence of selection bias associated with the inclusion of only patients scheduled for CABG in the present study. The results might differ in patients with less advanced atherosclerosis that do not require surgical treatment. However, this limitation could not be circumvented. The only solution to evade this bias in future is to identify the best biomarker in peripheral blood that reflects the cellular composition of the thymus. The evaluation of TREC concentration [69] might be one of the possible solutions, even though it also has certain shortcomings. Another important limitation of the study is that many patients exhibited a mixed phenotype of vascular aging, having both advanced atherosclerosis and high VAI values. This should be addressed in future studies by recruiting patients with intact coronary arteries but advanced aging of major blood vessels. Also, vascular aging was evaluated only via calculation of the VAI based on the data of ultrasound and arteriography. There are other approaches to measure vascular age, including those that are based on the determination of coronary artery calcium [5], as mentioned above. The possibility remains that other surrogate indices of vascular aging will be characterized by distinct associations with immune and molecular biomarkers, which warrants future studies.

Identification of distinct cellular subsets and features of thymus dysfunction involved in the manifestation of certain aging phenotypes open the road to the development of new therapeutic approaches targeting cardiovascular health in elderly patients. Current targets in inflammaging include cytokines (TNF, IL-6, and IL-1β), mitochondria, oxidative stress, various epigenetic markers, and the intestinal microbiome [10,70]. Evaluation of MDSC subsets and Treg features could improve risk stratification in CAD patients, while management of “thymflammation” and amendment of both adaptive and innate regulatory cells could be potential treatment options in the near future.

## 5. Conclusions

The most advanced and wide-spread atherosclerosis (GS ≥ 65 points) is associated with pro-inflammatory changes in the thymus, such as an increase in cytotoxic cells (CD8+ T lymphocytes and NK cells), a decrease in CD4 + CD8+ cells, and the absence of newly produced Treg lymphocytes. At the same time, thymic morphology remains comparable and ranges from total adipose substitution to 80% preservation of the thymus tissue. Changes in the peripheral blood include the decreased counts of eMDSCs, EPCs, and Treg lymphocytes, including cells with FoxP3 nuclear translocation. Advancement of coronary atherosclerosis and vascular aging apparently represent distinct phenotypes of biological aging associated with the opposite input of immune cells. Thus, MDSCs and Treg lymphocytes are equally involved in the progression of coronary atherosclerosis, which is aggravated by the decreased regulatory potential of the endothelium. Vascular aging represents a distinct phenotype of biological aging in CAD patients, which is characterized by the expansion of eMDSCs.

## Figures and Tables

**Figure 1 diagnostics-15-02494-f001:**
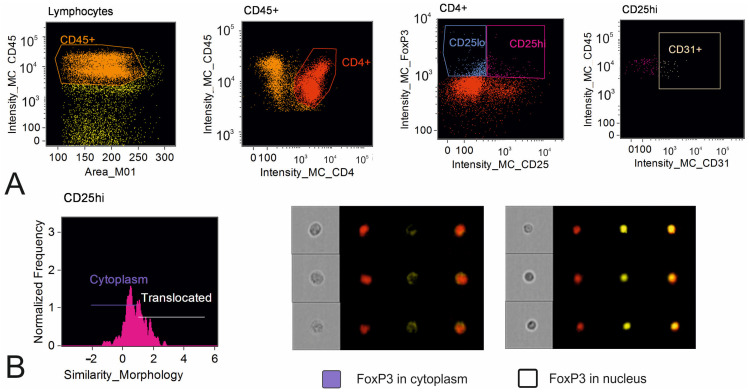
Analysis of FoxP3+ T regulatory lymphocytes in the thymus by imaging flow cytometry. (**A**) Gating strategy of FoxP3+ Tregs: identification of CD45 + cells (orange); identification of CD4+ (red); identification of CD25^hi^FoxP3+ (pink) and CD25^low^FoxP3+ Treg (blue) subsets; identification of recent thymic migrants: CD4 + FoxP3 + CD31+ cells (beige). (**B**) Analysis of FoxP3 nuclear translocation: histogram allowing identification of FoxP3 in the nucleus via the feature “Similarity Morphology” for the DNA stain (7AAD) and translocated marker (FoxP3) and representative cell images for cells with FoxP3 located in the cytoplasm and FoxP3 translocated to the nucleus; violet color in the graph indicates the region specific for FoxP3 in the cytoplasm; white color in the graph indicates the region specific for FoxP3 in the nucleus.

**Figure 2 diagnostics-15-02494-f002:**
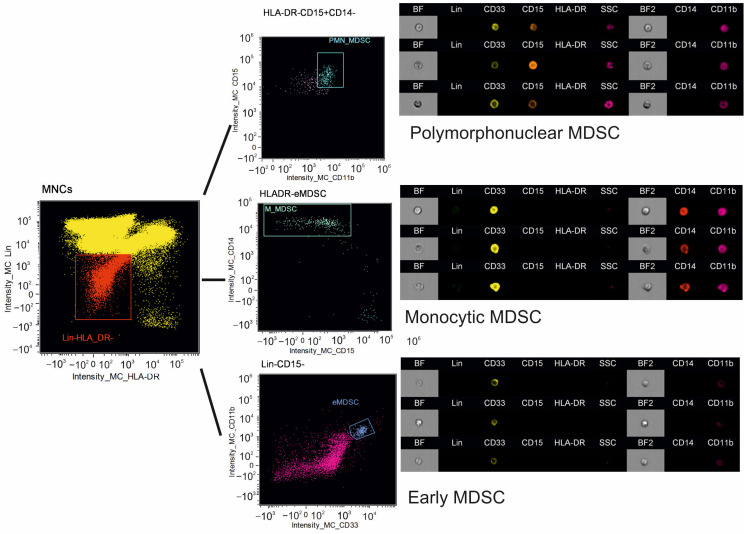
Gating strategy of myeloid-derived suppressor cells. In the population of Lin–HLA-DR– (red) cells, we identified CD33 + CD11b + CD15 + CD14– polymorphonuclear myeloid-derived suppressor cells (PMN-MDSCs) (turquoise), CD33 + CD11b + CD15–CD14+ monocytic MDSCs (M-MDSCs) (green), and CD33 + CD11b + CD15–CD14– early MDSCs (eMDSCs) (violet).

**Figure 3 diagnostics-15-02494-f003:**
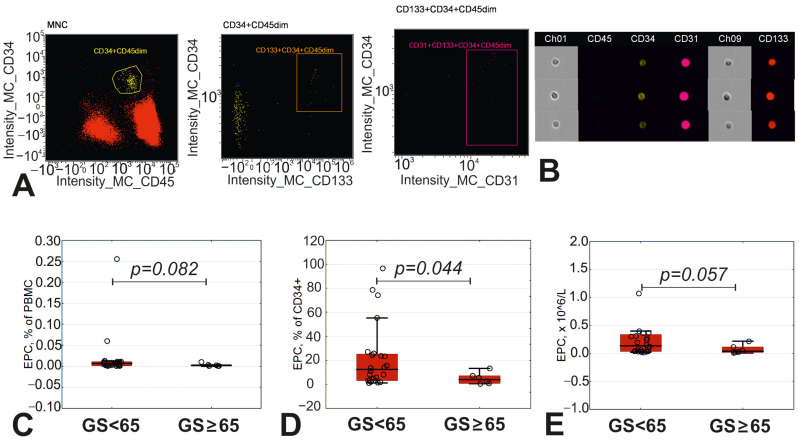
Endothelial precursor cells in patients with varying degrees of atherosclerosis. (**A**) Gating strategy of endothelial precursor cells (EPCs): in the population of mononuclear cells (MNCs) we gated CD34+CD45dim cells (yellow), followed by gating of CD133+CD34+CD45dim (orange), followed by gating of CD31+CD133+CD34+CD45dim cells (pink). (**B**) Representative images from image gallery for EPCs. (**C**) Percentage of EPCs in peripheral blood mononuclear cells (PBMCs). (**D**) Percentage of EPCs in CD34+ cells. (**E**) Absolute count of EPCs in peripheral blood.

**Figure 4 diagnostics-15-02494-f004:**
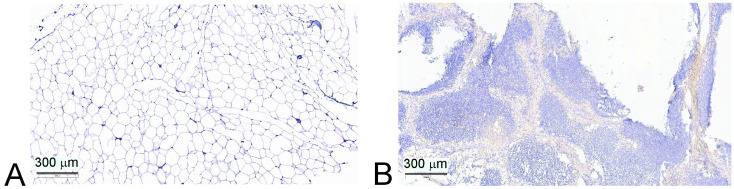
Photomicrographs of FoxP3 expression in the thymus. Scale bar: 300 μm. (**A**) Thymus is replaced with adipose tissue; no FoxP3+ expression in tissue. (**B**) Thymic morphology is preserved; FoxP3+ cells are present.

**Figure 5 diagnostics-15-02494-f005:**
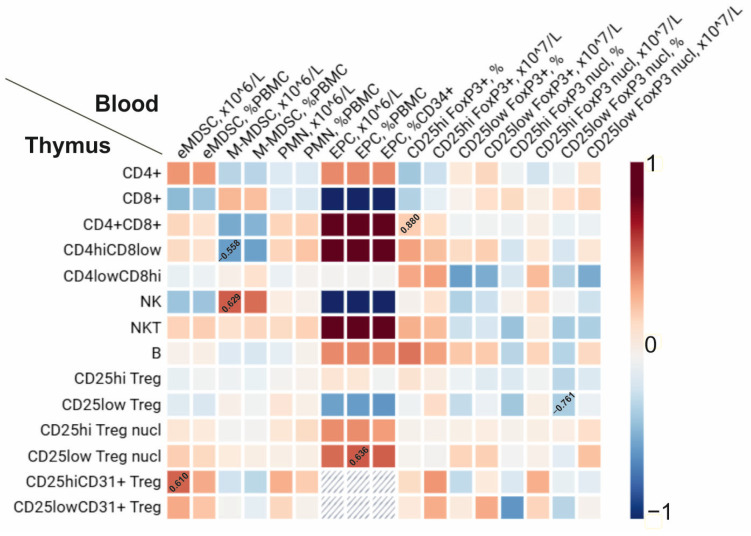
Correlations between immune cells in the thymus and those in peripheral blood. Positive correlations are marked in red; negative correlations are marked in blue; coefficients r_s_ are indicated only for significant correlations with *p* < 0.05.

**Figure 6 diagnostics-15-02494-f006:**
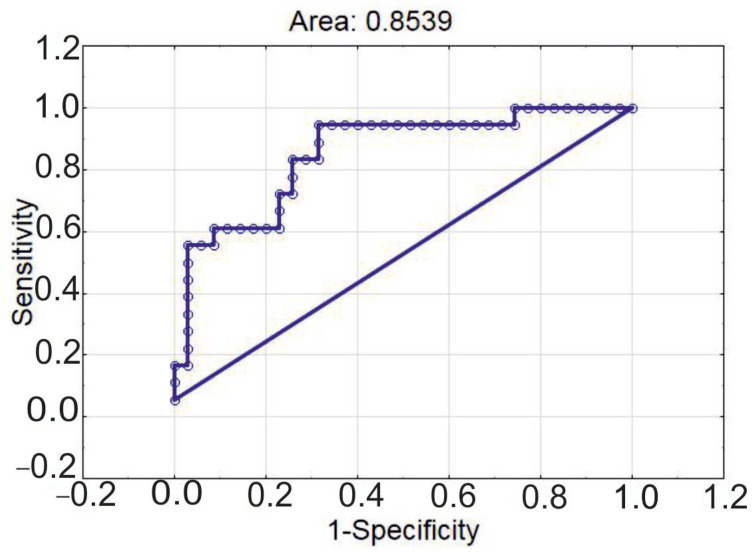
ROC-curve of multiple logistic regression for classification of patients into groups with varying degrees of atherosclerosis according to Gensini score.

**Figure 7 diagnostics-15-02494-f007:**
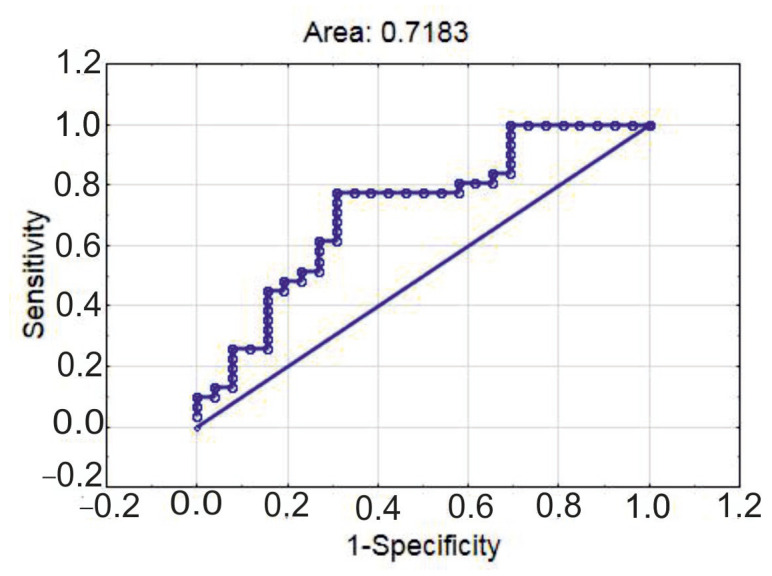
ROC-curve of the multiple logistic regression for the classification of patients into groups with a VAI below or above 83.9 points.

**Table 1 diagnostics-15-02494-t001:** Baseline characteristics of patients recruited to the study depending on atherosclerosis severity.

Characteristics	CAD GS < 65 (*n* = 37)	CAD GS ≥ 65 (*n* = 19)	*p*
Gender (m), *n* (%)	29 (78.4)	18 (94.7)	0.146
Age, years	65 (58; 69)	65 (58; 68)	0.918
History of myocardial infarction, *n* (%)	13 (35.1)	16 (84.2)	<0.001
Arterial hypertension, *n* (%)	34 (91.9)	19 (100)	0.544
Diabetes mellitus type 2, *n* (%)	13 (35)	7 (36.8)	0.999
Duration of coronary artery disease, years	2.5 (1.0; 9.0)	3.0 (1.0; 12.0)	0.428
Systolic blood pressure, mm Hg	125 (118; 137)	119 (112; 132)	0.066
Diastolic blood pressure, mm Hg	70 (65; 77)	70 (67; 76)	0.849
Smoking, *n* (%)	15 (40.5)	6 (31.6)	0.572
Body mass index, kg/m^2^	28.7 (26.4; 31.5)	28.7 (27.0; 31.8)	0.693
Waist circumference, cm	104 (97; 112)	103 (97; 111)	0.963
Gensini score, points	42.5 (12.5; 52)	77.0 (71.0; 101.5)	<0.001
cfPWV, m/s	10.0 (9.2; 11.2)	9.7 (9.3; 10.0)	0.139
IMT, mm	0.85 (0.75; 0.95)	0.87 (0.75; 1.00)	0.837
Vascular aging index	84.6 (79.5; 91.4)	82.8 (77.7; 89.69)	0.484
Oral hypoglycemic drugs, *n* (%)	7 (18.9)	7 (36.8)	0.195
Insulin, *n* (%)	4 (10.8)	1 (5.26)	0.652
RAAS inhibitors, *n* (%)	26 (70.3)	11 (57.9)	0.550
Calcium channels antagonists, *n* (%)	19 (51.4)	8 (42.1)	0.580
Beta-blockers, *n* (%)	30 (81.1)	15 (78.9)	0.999
Diuretics, *n* (%)	13 (35.1)	9 (47.4)	0.401
Statins, *n* (%)	34 (91.9)	18 (94.7)	0.998

CAD, coronary artery disease; cfPWV, carotid–femoral pulse wave velocity; GS, Gensini score; IMT, intima–media thickness; RAAS, renin–angiotensin–aldosterone system.

**Table 2 diagnostics-15-02494-t002:** Subsets of MDSC and Treg lymphocytes in peripheral blood.

Parameters	CAD GS < 65 (*n* = 37)	CAD GS ≥ 65 (*n* = 19)	*p*
PMN-MDSC, %	0.03 (0.02; 0.05)	0.03 (0.02; 0.05)	0.888
PMN-MDSC, ×10^6^/L	0.79 (0.47; 1.46)	0.64 (0.28; 1.17)	0.464
M-MDSC, %	0.11 (0.04; 0.47)	0.11 (0.06; 0.20)	0.845
M-MDSC, ×10^6^/L	3.01 (1.22; 10.96)	2.85 (1.18; 5.69)	0.706
eMDSC, %	0.31 (0.17; 0.61)	0.22 (0.10; 0.32)	0.154
eMDSC, ×10^6^/L	8.24 (5.08; 14.23)	4.02 (2.6 2; 8.70)	0.041
CD4 + CD25_hi_FoxP3+, %	6.18 (5.08; 7.44)	5.72 (4.57; 6.69)	0.166
CD4 + CD25^low^FoxP3+, %	1.19 (0.70; 1.74)	0.91 (0.74; 1.23)	0.097
CD4 + CD25^hi^FoxP3+, ×10^7^/L	5.91 (4.41; 8.67)	3.99 (3.38; 6.56)	0.049
CD4 + CD25^low^FoxP3+, ×10^7^/L	1.15 (0.69; 1.53)	0.76 (0.50; 0.94)	0.007
CD25^hi^ FoxP3 nuclear, %	96.5 (84.1; 98.1)	89.5 (73.1; 97.3)	0.191
CD25^low^ FoxP3 nuclear, %	89.7 (76.5; 93.6)	76.2 (63.8; 96.8)	0.312
CD25^hi^ FoxP3 nuclear, ×10^7^/L	5.20 (3.97; 7.47)	3.73 (2.30; 6.00)	0.031
CD25^low^ FoxP3 nuclear, ×10^7^/L	0.91 (0.59; 1.26)	0.56 (0.31; 0.85)	0.009

PMN-MDSC, polymorphonuclear myeloid derived suppressor cell; M-MDSC, monocytic myeloid derived suppressor cell; eMDSC, early myeloid derived suppressor cell; for MDSC subsets, the percentage is related to the total number of PBMC cells; for Treg cells, the percentage is related to the total number of CD4+ cells; heading indexed as “nuclear” indicates the number of Treg cells with FoxP3 nuclear translocation, and represents the percentage of the total number of the FoxP3+ Treg cells.

**Table 3 diagnostics-15-02494-t003:** Results of the immunohistochemical analysis of the thymus.

Parameters	CAD GS < 65 (*n* = 15)	CAD GS ≥ 65 (*n* = 7)	*p*
Preserved thymic parenchyma, %	2 (1; 10)	2 (0; 15)	0.636
FoxP3+ cells, *n* (%)	6 (40.0)	2 (28.5)	0.999

**Table 4 diagnostics-15-02494-t004:** T-, B-, and NK lymphocytes and subsets of Treg lymphocytes in thymic stromal–vascular fraction.

Parameters	CAD GS < 65 (*n* = 17)	CAD GS ≥ 65 (*n* = 11)	*p*
CD4+ T lymphocytes, %	33.6 (30.5; 45.5)	29.6 (25.4; 31.6)	0.113
CD8+ T lymphocytes, %	19.5 (15.8; 27.2)	30.3 (29.9; 31.0)	0.026
CD4 + CD8+ T lymphocytes, %	1.3 (0.7; 2.4)	0.3 (0.2; 0.6)	0.036
CD4^hi^CD8^low^ T lymphocytes, %	0.5 (0.3; 1.2)	0.2 (0.1; 0.4)	0.067
CD4^low^CD8^hi^ T lymphocytes, %	0.3 (0.2; 0.4)	0.2 (0; 0.5)	0.529
NK cells, %	3.1 (1.8; 6.0)	14.2 (10.9; 16.7)	0.018
NKT cells, %	9.9 (4.7; 14.6)	10.9 (5.8; 18.9)	0.607
B lymphocytes, %	5.7 (3.8; 8.5)	3.3 (2.6; 4.2)	0.328
CD4 + CD25^hi^FoxP3+, %	7.5 (5.6; 13.9)	7.0 (4.7; 16.3)	0.853
CD4 + CD25^low^FoxP3+, %	3.1 (2.2; 6.6)	4.5 (1.0; 6.8)	0.735
CD25^hi^ FoxP3 nuclear, %	33.1 (22.6; 44.5)	19.4 (12.5; 26.0)	0.134
CD25^low^ FoxP3 nuclear, %	20.2 (7.3; 28.8)	10.8 (0; 18.7)	0.122
CD4 + CD25^hi^FoxP3 + CD31+, %	1.7 (0.3; 11.4)	0	0.085
CD4 + CD25^low^FoxP3 + CD31+, %	0 (0; 1.2)	0	0.517

For T-, B-, and NK cells, the percentage is related to the total number of CD45+ lymphocytes in the stromal–vascular fraction; for Treg cells, the percentage is related to the total number of CD4+ cells.

**Table 5 diagnostics-15-02494-t005:** Senescence-associated molecular markers in patients depending on the degree of atherosclerosis.

Parameters	CAD GS < 65 (*n* = 37)	CAD GS ≥ 65 (*n* = 19)	*p*
IL-1β, pg/mL	0.82 (0.54; 1.17)	0.79 (0.51; 1.15)	0.887
IL-10, pg/mL	1.93 (1.53; 3.21)	2.34 (1.66; 2.63)	0.745
IL-6, pg/mL	1.53 (0.87; 2.39)	1.50 (1.27; 1.87)	0.530
TNF, pg/mL	2.02 (0.71; 2.78)	2.38 (1.46; 2.62)	0.652
TGFβ, ng/mL	32.08 (25.62; 38.34)	32.57 (19.05; 37.17)	0.562
IGF, μg/mL	77.27 (58.26; 106.00)	66.11 (42.59; 100.00)	0.411
Homocysteine, μmol/L	11.51 (9.91; 14.75)	12.10 (9.73; 14.26)	0.951
Endocan-1, pg/mL	1893.5 (1518.5; 2571.0)	2304.0 (1922.0; 2603.0)	0.114
Endothelin-1, fmol/mL	0.38 (0.31; 0.78)	0.44 (0.37; 0.56)	0.568
Resistin, ng/mL	4.71 (3.85; 6.33)	4.40 (3.76; 5.62)	0.540
Sortilin, ng/mL	11.73 (4.54; 116.08)	9.37 (5.44; 36.21)	0.835

**Table 6 diagnostics-15-02494-t006:** The model of logistic regression for the prediction of the Gensini score increase above 65 points.

	Estimates	95% CI (Lower)	95% CI (Upper)	*p*
Intercept	−0.678	0.05938	4.34305	0.536
Infarction	2.752	2.85114	86.24346	0.002
eMDSC, ×10^6^/L	−0.081	0.82418	1.03149	0.922
CD25^low^ FoxP3 nuclear, ×10^7^/L	−1.514	0.02624	1.84440	0.163

**Table 7 diagnostics-15-02494-t007:** The model of logistic regression to predict the increase in VAI above its median value.

	Estimates	95% CI (Lower)	95% CI (Upper)	*p*
eMDSC, ×10^6^/L	1.104639	1.001398	1.218523	0.047
age	1.050619	0.984182	1.121541	0.138

## Data Availability

The raw data supporting the conclusions of this article will be made available by the authors on request.

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
