# Peer review of "Associations Between Regulatory Immune Cells, Thymus Cellular Remodeling, and Vascular Aging in Advanced Coronary Atherosclerosis: A Pilot Study"

_diagnostics, 2025, doi:10.3390/diagnostics15192494_

Round 1
Reviewer 1 Report
Comments and Suggestions for Authors
The study topic is original and clinically relevant, linking immune aging, thymus function, and CAD progression. However, the abstract is quite dense, with long sentences that obscure the main findings.
The introduction should more clearly define why exploring thymus morphology and regulatory immune cells in CAD is important (i.e., novelty vs prior work). The current framing mixes vascular aging, immunosenescence, and CAD without a smooth narrative. A short paragraph explicitly stating the knowledge gap would strengthen the rationale.
Findings are interesting, but currently overcrowded in long sentences. Consider breaking them into separate points.
The discussion should emphasize the novelty: thymus dysfunction in CAD relates more to cellular composition than structural involution.
Author Response
Dear Reviewer,
Thank you for your time and efforts that you spent working on our submission. We tried our best to improve the shortcomings that you identified. All the corrections, except small editions of the text are marked with blue color. Below we represent the summary of all the corrections that were introduced.
Reviewer #1
The study topic is original and clinically relevant, linking immune aging, thymus function, and CAD progression. However, the abstract is quite dense, with long sentences that obscure the main findings.
Thank you, we tried to make abstract more concise and tried to outline the main findings of the study.
The introduction should more clearly define why exploring thymus morphology and regulatory immune cells in CAD is important (i.e., novelty vs prior work). The current framing mixes vascular aging, immunosenescence, and CAD without a smooth narrative. A short paragraph explicitly stating the knowledge gap would strengthen the rationale.
Thank you for your suggestion. We introduced connecting sentences between the paragraphs. We tried to simplify the sentences, and added justification of the research in introduction, stating the knowledge gap and practical significance of the study.
Findings are interesting, but currently overcrowded in long sentences. Consider breaking them into separate points.
Thank you for pointing this out. We tried to simplify the narrative in the Results section.
The discussion should emphasize the novelty: thymus dysfunction in CAD relates more to cellular composition than structural involution.
We tried to address this point and edited the Discussion section. We also tried to simplify sentences and make them more understandable.
Reviewer 2 Report
Comments and Suggestions for Authors
This study investigates how immune regulation and thymus remodeling contribute to aging-related vascular changes in coronary artery disease (CAD). Patients with severe atherosclerosis (Gensini Score ≥65) exhibited reduced early myeloid-derived suppressor cells (eMDSC), endothelial progenitor cells (EPC), and FoxP3+ regulatory T cells, alongside impaired thymic output of new Tregs and increased cytotoxic CD8+ and NK cells. Interestingly, thymus morphology varied but did not correlate directly with disease severity; instead, cellular composition—particularly the absence of CD31+ recent thymic emigrants—was more telling. The findings suggest that coronary atherosclerosis and vascular aging are distinct immunological aging phenotypes: the former marked by immune regulatory cell depletion and inflammation, the latter by eMDSC expansion. These immune shifts may undermine vascular repair and promote disease progression. Specific comments:
- The paper distinguishes between coronary atherosclerosis and vascular aging as separate aging phenotypes. Could the authors clarify whether these are mutually exclusive or overlapping in clinical presentation, and how this distinction informs therapeutic strategies?
- With only 19 patients in the GS≥65 group and 28 thymus biopsies total, how was statistical power calculated for subgroup analyses, especially those involving thymic cellular composition?
- Since thymus samples were only obtained during CABG, how do the authors account for potential selection bias? Might these patients differ systematically from those not undergoing surgery?
- The study relies heavily on phenotypic markers and nuclear translocation of FoxP3. Were any functional assays (e.g., suppression assays) performed to validate the immunosuppressive capacity of these cells?
- Given the cross-sectional design, how do the authors address the possibility that immune cell alterations are consequences rather than drivers of atherosclerosis or vascular aging?
- The absence of CD31+ Tregs in advanced CAD patients is interpreted as lack of thymic output. Could alternative explanations—such as altered trafficking or peripheral consumption—be considered?
- Although SASP markers did not differ significantly between groups, several correlations were reported. How do the authors reconcile this with the broader literature on SASP as a hallmark of vascular aging?
- The VAI is derived from carotid-femoral pulse wave velocity and intima-media thickness. Would inclusion of coronary calcium scores or other imaging modalities strengthen the vascular aging assessment?
- The increase in cytotoxic cells within the thymus is intriguing. Could this reflect peripheral recirculation rather than intrinsic thymic remodeling? Were markers of thymic residency or activation assessed?
- The study suggests immune profiling could stratify aging phenotypes in CAD. What are the practical implications for patient management, and could these biomarkers guide immunomodulatory interventions?
Author Response
Dear Reviewer,
Thank you for your time and efforts that you spent working on our submission. We tried our best to improve the shortcomings that you identified. All the corrections, except small editions of the text are marked with blue color. Below we represent the summary of all the corrections that were introduced.
Reviewer #2
- The paper distinguishes between coronary atherosclerosis and vascular aging as separate aging phenotypes. Could the authors clarify whether these are mutually exclusive or overlapping in clinical presentation, and how this distinction informs therapeutic strategies?
Thank you for pointing this out. Vascular aging may either precede the development of atherosclerosis and persist in CAD patients together with the plaques or represent a distinct aging phenotype. Identification of pathological factors that determine increase of arterial stiffness and vascular aging, and lead to its progression to atherosclerosis may help in prevention of unfavorable cardiovascular events and target organ damage. – We added this information in Introduction.
- With only 19 patients in the GS≥65 group and 28 thymus biopsies total, how was statistical power calculated for subgroup analyses, especially those involving thymic cellular composition?
We acknowledge that being a pilot study, our research does not have a high statistical power, which we mentioned as one of the limitations. We underscored its pilot nature by modification of the title.
- Since thymus samples were only obtained during CABG, how do the authors account for potential selection bias? Might these patients differ systematically from those not undergoing surgery?
Indeed, we could not evade the potential selection bias. We have mentioned it in limitations of the study.
- The study relies heavily on phenotypic markers and nuclear translocation of FoxP3. Were any functional assays (e.g., suppression assays) performed to validate the immunosuppressive capacity of these cells?
Unfortunately, we did not have possibility to perform functional tests in this study, but according to the previous works of other researchers, FoxP3 subcellular localization strongly correlates with Treg functional properties evaluated by various approaches. We have added this into discussion section (doi: 10.1002/eji.201141838; doi: 10.15252/embj.201899766).
Also, previously we published our results on the ability of thymus stromal –vascular fraction cells to produce cytokines, and its correlation with FoxP3 nuclear translocation. We added this information, as well (doi: 10.1007/s10517-025-06304-2).
- Given the cross-sectional design, how do the authors address the possibility that immune cell alterations are consequences rather than drivers of atherosclerosis or vascular aging?
Thank you for pointing this out. Indeed, in cross-sectional study the “cause and effect” relationships cannot be elucidated. We addressed this in limitations of the study.
- The absence of CD31+ Tregs in advanced CAD patients is interpreted as lack of thymic output. Could alternative explanations—such as altered trafficking or peripheral consumption—be considered?
CD31 molecule is expressed on the naïve T cells that are just leaving thymus. So, absence of CD31+ cells in thymus indicates that no newly Tregs are present in patients with advanced coronary atherosclerosis. The problem remains, that this marker may be not the most specific one. But the low specificity would have accounted for the presence of CD31+ cells, which are not recent thymic emigrants in reality. Thus, we believe that absence of CD31+ Tregs in thymus may be regarded as a lack of thymic output. In the future the results may be corroborated by expansion of fluorochromes panel or TREC detection. We did not have such technical opportunities at present, but added this information in discussion (doi: 10.1016/j.bbcan.2022.188730). And hope to work on this problem as soon as we get access to equipment, allowing to expand our fluorochrome panel.
- Although SASP markers did not differ significantly between groups, several correlations were reported. How do the authors reconcile this with the broader literature on SASP as a hallmark of vascular aging?
Most likely absence of interconnections between SASP markers and severity of CAD may be explained by the fact that both groups of patients had rather sever atherosclerosis, as all the patients required surgical correction of stenosis via CABG. Thus, both groups had comparable levels of SASP markers. Also, serum SASP concentrations might not be very representative for the local processes with the plaque, and subtle changes could not be captured, as many SASP effects are produced through paracrine interactions (doi: 10.1186/s13578-022-00815-5). We added this explanation in discussion.
- The VAI is derived from carotid-femoral pulse wave velocity and intima-media thickness. Would inclusion of coronary calcium scores or other imaging modalities strengthen the vascular aging assessment?
Indeed, coronary artery calcium evaluation might improve evaluation of vascular age. We have mentioned it in the concluding remarks of the discussion among the other limitations, but reinforced this point even more in the discussion (doi: 10.1002/clc.23955; doi: 10.1080/02813432.2025.2456948; doi: 10.1093/eurjpc/zwad134).
- The increase in cytotoxic cells within the thymus is intriguing. Could this reflect peripheral recirculation rather than intrinsic thymic remodeling? Were markers of thymic residency or activation assessed?
Unfortunately, we did not have possibility to evaluate markers of thymic residency in effector T, B NK cells. We cannot exclude that these cells (at least in part) recirculated to thymus. However, the presence of effector lymphoid cells in thymus was demonstrated in other studies before. They appeared to be obligatory in normal thymus to establish tolerance to self-molecules (doi: 10.1111/imr.70037). However, function of these cells in atrophied thymus is unknown yet. We added this into discussion.
- The study suggests immune profiling could stratify aging phenotypes in CAD. What are the practical implications for patient management, and could these biomarkers guide immunomodulatory interventions?
Thank you, indeed, this information should have been included in Discussion to address practical significance of the study. To address possible implications of the obtained results we added a paragraph connecting Discussion and Conclusion sections (doi: 10.3390/cells11061010; doi: 10.1093/cvr/cvae240).
Round 2
Reviewer 2 Report
Comments and Suggestions for Authors
This study investigates how immune regulation and thymus remodeling contribute to aging-related vascular changes in coronary artery disease (CAD). Patients with severe atherosclerosis (Gensini Score ≥65) exhibited reduced early myeloid-derived suppressor cells (eMDSC), endothelial progenitor cells (EPC), and FoxP3+ regulatory T cells, alongside impaired thymic output of new Tregs and increased cytotoxic CD8+ and NK cells. Interestingly, thymus morphology varied but did not correlate directly with disease severity; instead, cellular composition—particularly the absence of CD31+ recent thymic emigrants—was more telling. The findings suggest that coronary atherosclerosis and vascular aging are distinct immunological aging phenotypes: the former marked by immune regulatory cell depletion and inflammation, the latter by eMDSC expansion. The revision of the manuscript is much improved, no additional comments.